# Interactions between Quantum Dots and G-Actin

**DOI:** 10.3390/ijms241914760

**Published:** 2023-09-29

**Authors:** Nhi Le, Abhishu Chand, Emma Braun, Chloe Keyes, Qihua Wu, Kyoungtae Kim

**Affiliations:** 1Department of Biology, Missouri State University, Springfield, MO 65897, USA; nhi0407@live.missouristate.edu (N.L.); ac43s@missouristate.edu (A.C.); eb2872s@missouristate.edu (E.B.); 2Jordan Valley Innovation Center, Springfield, MO 65806, USA; chloekeyes@missouristate.edu (C.K.); qwu@missouristate.edu (Q.W.)

**Keywords:** quantum dots, actin, interaction, fluorescence quenching, toxicity, secondary structure

## Abstract

Quantum dots (QDs) are a type of nanoparticle with excellent optical properties, suitable for many optical-based biomedical applications. However, the potential of quantum dots to be used in clinical settings is limited by their toxicity. As such, much effort has been invested to examine the mechanism of QDs’ toxicity. Yet, the current literature mainly focuses on ROS- and apoptosis-mediated cell death induced by QDs, which overlooks other aspects of QDs’ toxicity. Thus, our study aimed to provide another way by which QDs negatively impact cellular processes by investigating the possibility of protein structure and function modification upon direct interaction. Through shotgun proteomics, we identified a number of QD-binding proteins, which are functionally associated with essential cellular processes and components, such as transcription, translation, vesicular trafficking, and the actin cytoskeleton. Among these proteins, we chose to closely examine the interaction between quantum dots and actin, as actin is one of the most abundant proteins in cells and plays crucial roles in cellular processes and structural maintenance. We found that CdSe/ZnS QDs spontaneously bind to G-actin in vitro, causing a static quenching of G-actin’s intrinsic fluorescence. Furthermore, we found that this interaction favors the formation of a QD–actin complex with a binding ratio of 1:2.5. Finally, we also found that CdSe/ZnS QDs alter the secondary structure of G-actin, which may affect G-actin’s function and properties. Overall, our study provides an in-depth mechanistic examination of the impact of CdSe/ZnS QDs on G-actin, proposing that direct interaction is another aspect of QDs’ toxicity.

## 1. Introduction

For the past few decades, nanoparticles have attracted worldwide attention for their potential to revolutionize modern science and technology. In particular, the use of nanoparticles in biomedical science has shown promising results. Among these nanoparticles, quantum dots (QDs) stood out as a shining candidate due to their many unique characteristics. QDs are nano-sized semiconductor crystals with the ability to emit a broad range of bright, photobleaching-resistant fluorescence [1,2]. QDs often possess an encapsulating shell with conjugated surface ligands, which help them remain relatively soluble and stable under biological conditions [3,4,5,6,7]. These features have made QDs highly useful for optical detection-based biomedical applications [8,9,10,11,12,13,14,15,16,17]. However, recent studies have reported the toxicity of QDs toward cells [18,19,20,21,22,23,24,25,26], causing many to raise concerns regarding the use of QDs in biological settings. Consequently, much effort has been invested to evaluating the safety of QDs and providing strategies to improve QDs’ biocompatibility. The toxicity of QDs has been shown to be complex, as it is influenced by a number of factors such as QDs’ shape, size, composition, and ligand types [27,28,29,30,31,32,33,34]. Yet, most current studies on QDs primarily focus on apoptotic-based toxicity [35,36], thus overlooking the potential of other factors that may contribute to QDs’ toxicity.

With their easily detectible fluorescence and ability to be taken up by cells [37,38,39], quantum dots are considered a great material for applications such as drug delivery [11,40,41,42]. In QD-based drug delivery systems, the role of ligands on the surface of QDs is crucial. Using the correct ligand could increase the stability of the drug delivery vehicle by preventing the premature release of loaded drugs as well as act as a homing peptide to guide the delivery vehicle to the correct target site [43,44,45,46]. Once entered in the cells and the treatments are released, exposed ligands and functional groups on QDs are free to have non-specific interactions with cellular proteins. Furthermore, the formation of a “protein corona” on the surface of QDs has previously been reported. The interaction between nanoparticles and proteins can cause severe protein modifications that affect the function and properties of proteins [47,48,49]. Coronation of protein on the nanoparticle’s surface could also limit the supply of protein that can carry out important biological processes, leading to a serious adverse effect on cells. While most of the research regarding the interaction of QDs with proteins has only used common proteins found in serum such as human serum albumin (HSA) or bovine serum albumin (BSA), research on QDs’ interaction with intracellular proteins is very sparse. As such, the present research aims to look for intracellular proteins that could interact with cadmium selenide zinc sulfide quantum dots (CdSe/ZnS QDs) and characterize their interactions. In this paper, we used proteomics shotgun analysis to identify several intracellular proteins that were pulled down by CdSe/ZnS QDs. Then, choosing actin as a representative QD-binding intracellular protein, we used an array of biochemical methodologies to reveal the mode of interaction between QDs and actin. Finally, we used circular dichroism spectroscopy to assess if CdSe/ZnS QDs could alter the structure of G-actin upon direct interaction.

## 2. Results

### 2.1. Identification of QD Binding Protein

Proteins are an essential component of cells. It is well known that the structure of proteins is critical for their function and property [50]. Thus, alteration of protein structure resulting from non-specific interaction with nanoparticles could lead to unfavorable consequences. As most of the recently developed QD-based biomedical technologies, such as drug delivery, have involved the internalization of QDs into cells, we decided to investigate the interaction between quantum dots and intracellular proteins. We incubated protein lysate with CdSe/ZnS QDs and then identified QD-binding proteins using mass spectrometry. Our results revealed several QD-binding proteins. These proteins are associated with a number of important biological processes, including translation, transcription, heat shock, mitochondrial structure and function, vesicular trafficking, and the actin cytoskeleton (Figure 1). Our result suggested that QDs are capable of non-specific interaction with intracellular proteins. To further investigate the interaction between QDs and proteins, we performed a series of biochemical experiments to characterize the binding mechanism between QDs and actin proteins. We chose to work with actin for several reasons. The actin cytoskeleton plays an important role in many cellular processes including nutrient uptake, vesicle transport, and cellular structure maintenance [51,52,53,54,55]. However, it was previously reported that treatment of CdSe/ZnS QDs caused abnormal appearance in yeast actin filament [39]. Furthermore, the reason behind this abnormal appearance has been largely unexplored. Thus, we wanted to look closely at the interaction between QDs and actin to assess if this interaction resulted in an alteration in actin structure and function.

### 2.2. Validation of QD–Actin Interaction via a Native Gel Analysis

To verify the binding of G-actin to QDs, we performed native polyacrylamide gel electrophoresis (PAGE), a method often used in studies to separate proteins in their native form by size and charge. Our G-actin had a smaller molecular weight of 43 kDa, allowing it to move faster through the gel compared to the QDs, which were around 1350 kDa. If G-actin did indeed bind to QDs, we would see a retardation of the G-actin protein band on the native gel. The binding ratio of QDs and G-actin could then be estimated when all G-actin was retarded by QDs. To examine the binding ratio between the QDs and G-actin, we created a series of samples containing a fixed concentration of G-actin incubating with a gradient in concentrations of QDs. For each quantum dot concentration tested, we also created a QD-alone sample as a reference for comparison. The result from our gel showed a darkening of QD bands in QD–actin samples compared to the bands of the QD-alone samples with the corresponding QDs’ concentration (Figure 2A,B). Thus, this result showed that G-actin was able to bind to quantum dots and was retained at the QD bands instead of moving downward on the gel. In addition, the difference in QD band intensity between the QD–actin samples and the QD-alone sample gradually increased with increasing QD concentration, reaching a maximum at 0.8 µM QDs (average intensity difference: 24.033 a.u., STDEV 4.9), then decreasing with higher concentration (Figure 2B). Furthermore, the G-actin band was no longer observed for a QD concentration of 0.8 µM and higher. This suggests that 0.8 µM of QDs was the optimal concentration to bind 2 µM of G-actin. Thus, the binding ratio of QDs to G-actin was found to be 1 to 2.5.

To further verify our results, we tested samples with a fixed amount of QDs (0.2 µM) and an increasing concentration of actin. If our previous results were accurate, the binding ratio would remain in proximity. Consistently, our gel showed higher intensities at the QD bands as G-actin concentration increased (Figure 2C,D). However, samples with actin concentrations higher than 0.4 µM had observable free actin bands, indicating that 0.2 µM of QDs could not cause complete retardation of G-actin with concentrations higher than 0.4 µM. Thus, the binding ratio of QDs to G-actin in this case was determined to be 1 to 2, which is relatively close to the binding ratio we obtained above.

In the literature, the affinity between two proteins is often expressed in terms of K_D_ (dissociation constant) values. Lower K_D_ values would indicate a higher affinity and vice versa. Similarly, we could also use K_D_ values to evaluate the strength of the interaction between QDs and G-actin. Using the method outlined by Chambers et al. [56], we estimated the K_D_ values for quantum dots to be around 400 nM (Figure 2D). So far, our data have supported the binding of QDs and G-actin. However, the specific mechanism in which they are bound and how this affects G-actin remains unknown. For the next portion of our study, we attempted to investigate the characteristics of the interaction between QDs and G-actin by using the study of thermodynamics.

### 2.3. The Quenching of G-Actin’s Intrinsic Fluorescence by QDs

Previously, it has been reported that proteins composed of certain amino acid residues such as phenylalanine, tryptophan, and tyrosine have an intrinsic fluorescence that can be detected upon excitation [57]. When these proteins interact with other biomolecules, a strong interaction may modify the position of these amino acids, resulting in the quenching of their intrinsic fluorescence. Tighter interactions cause more alterations to the position of these amino acids; thus, significant quenching of the protein’s intrinsic fluorescence suggests a stronger binding affinity [57]. Today, this fluorescence quenching technique coupled with thermodynamic principles is widely used to characterize the interaction between proteins. According to the current literature, the two common types of fluorescence quenching are dynamic quenching and static quenching [58,59]. In dynamic quenching, the interactions between proteins are collision-based [60,61]. As the collision between molecules happens more frequently with enhanced temperature, the effect of dynamic quenching would also increase with higher temperatures [62]. On the other hand, static quenching is associated with complex formation [63]. It is well known that high temperature tends to favor the disassociation of complexes. Therefore, the effect of static quenching would decrease with higher temperatures [64]. As G-actin contains multiple tryptophan and phenylalanine amino acids [65], using the fluorescence quenching technique, we decided to examine if the interaction between QDs and G-actin was due to spontaneous collision or static complex formation. We measured the intrinsic fluorescence of G-actin alone versus G-actin with QDs at three different temperatures. Our results showed that higher concentrations of quantum dots significantly quenched actin’s intrinsic fluorescence at all three temperatures, while lower concentrations had minimal impacts on the intrinsic fluorescence of actin (Figure 3A–C). The quenching of actin’s intrinsic fluorescence indicates that QDs can bind to G-actin and cause alteration in the position of amino acids such as tryptophan and phenylalanine. Furthermore, the quenching effect caused by 20.6 nM of QDs was the greatest at 295 K (average of 21.7 percent reduction), compared to 303 K (average of 16.39 percent reduction) and 310 K (average of 15.6 percent reduction). This indicates that the quenching effect was abated at a higher temperature, suggesting a static quenching mechanism.

Currently, the literature on CdSe/ZnS QDs has suggested that the negative impact of these QDs in cells may result from the leakage of cadmium ions [66,67]. Therefore, it was important to investigate if the fluorescence quenching that we observed was influenced by leaked cadmium (Cd^2+^) ions. To achieve this, we first examined the amount of Cd^2+^ ions leaked from QDs over the course of 14 days. According to our results, the amount of leaked Cd^2+^ ions from QDs after 14 days was below the detectible limit of 50 ppb or 50 ng/mL (Figure 3D). Based on this result, we performed a fluorescence quenching assay where we treated 5–100 ng/mL of Cd^2+^ ions in the form of CdSO_4_ and measured the intrinsic fluorescence of G-actin. Our data revealed that at 5–100 ng/mL of Cd^2+^ ions, the intrinsic fluorescence of G-actin did not quench the intrinsic fluorescence of G-actin (Figure 3E). Therefore, we concluded that the previously observed fluorescence quenching of G-actin by CdSe/ZnS QDs was not influenced by leaked cadmium ions.

Next, we used the Stern–Volmer equation (Equation (1)) to reconfirm if the quenching of actin intrinsic fluorescence is dynamic or static quenching.
(1)$$\frac{F_{0}}{F}=1+K_{\mathrm{SV}}[Q]$$

In this equation, F_0_ is the intrinsic fluorescence of actin without QDs, F is the intrinsic fluorescence of actin in the presence of QDs, [Q] is the concentration of QDs, and K_SV_ is the Stern–Volmer quenching constant that needs to be calculated. A higher K_SV_ indicates more fluorescence quenching. Thus, if the K_SV_ value increases with higher temperature, dynamic quenching is likely to be the quenching mechanism. On the other hand, if the K_SV_ values decrease as the temperature increases, this would indicate that static quenching is likely to be the reason. The slope of each line in Figure 4A represents the K_SV_ value at each temperature. According to our calculations, the K_SV_ values decreased as the temperature increased (Figure 4A and Table 1). Thus, our data suggest that the quenching of the intrinsic fluorescence of actin is mediated by static quenching.

As mentioned in Section 2.2, the affinity of protein interactions is often expressed in terms of the dissociation constant (K_D_). However, the affinity could also be expressed in terms of the association constant (K_a_). The higher the K_a_ value, the stronger the affinity. Thus, with our current fluorescence quenching data, we used a modified version of the Stern–Volmer equation (Equation (2)) to figure out the K_a_ value. In the modified Stern–Volmer equation,
(2)$$\frac{F_{0}}{F_{0}-\;F}=\frac{1}{f_{a}\;K_{a}}\frac{1}{\left[Q\right]}+\frac{1}{f_{a}}$$
where f_a_ represents the concentration of the quencher-accessible fluorophore (G-actin) and K_a_ is the effective quenching constant (association constant) that needs to be calculated. Based on our calculation, the K_a_ value decreased as the temperature increased (Figure 4B, Table 1), suggesting a weaker binding affinity between QDs and G-actin at higher temperatures. Thus, these data further support static quenching and complex formation for QD–actin interaction.

To further investigate if the interaction between QDs and G-actin was spontaneous or non-spontaneous, we used the van ’t Hoff equation (Equations (3) and (4)) to calculate the change in enthalpy (∆H), the change in entropy (∆S), and the change in Gibbs free energy (∆G). Using the van ’t Hoff equation,
(3)$$\mathrm{Ln}\;K_{a}=-\frac{∆H}{\mathrm{RT}}+\frac{∆S}{R}$$
∆G = ∆H − T∆S(4)
we found a negative ∆G (Table 1), indicating that the interaction between QDs and G-actin is spontaneous. In a spontaneous reaction, the formation of products is favored without an additional stimulus such as heat. Collectively, our data support QD–actin complex formation.

### 2.4. Complex Formation Assessment Using UV–Vis Absorption Spectroscopy

So far, our data consistently indicate that the interaction between actin and QDs favors the formation of a QD–actin complex. Thus, our next step was to confirm the formation of the QD–actin complex. To do this, we used UV–vis spectroscopy to measure the absorption of G-actin alone versus the absorption of actin when incubated with QDs. According to several papers that have employed similar methods [57,68], we could calculate the absorption of G-actin in the presence of QDs using the equation below (Equation (5)):[Calculated Actin abs.] = [Abs. of QD–actin mixture] − [Abs. of QDs alone](5)

Afterward, the calculated actin absorption was compared to the true absorbance of actin that was not incubated with quantum dots. Overlapping between the two absorbances would indicate that no complex formation had taken place. If the two absorbances did not overlap, this would confirm complex formation between the QDs and G-actin. Our results showed that there was a downward shift in the absorbance of G-actin (Figure 5). Therefore, our data confirm that the interaction between QDs and G-actin results in the formation of QD–actin complexes.

### 2.5. G-Actin Hydrodynamic Diameter Increased in the Presence of CdSe/ZnS QDs

Another method commonly used to measure binding between two objects is to measure the diameter using dynamic light scattering (DLS). An increase in size suggests the binding of the two particles. Thus, we investigated the change in the diameter of G-actin with different concentrations of quantum dots. We found that the diameter of actin without quantum dots was around 7.76 nm with a standard deviation of 0.283 nm. In the presence of QDs, the diameter of actin increased to 16.12 ± 1.089 nm for samples with 4.86 nM QDs, 14.04 ± 0.142 nm for samples with 19.44 nM QDs, 17.375 ± 2.3405 nm for samples with 38.8 nM QDs, and 17.14 ± 0.08 nm for samples with 48.60 nM QDs (Figure 5B), and the G-actin peak previously observed (7.76 ± 0.283 nm) was no longer detected for quantum dot-treated groups. This change in diameter indicates that the G-actin protein bound to quantum dots.

### 2.6. The Alteration of Actin’s Secondary Structure by CdSe/ZnS QDs

It has been reported that nanomaterials can alter the structure of proteins upon interaction [69]. Changes in the structure of a protein often lead to functional changes or impairment, which could significantly impact many cellular processes [50,70]. Thus, after conducting a series of experiments to characterize the interaction between quantum dots and G-actin, we wanted to know the impact of QDs on the structure of G-actin. To accomplish this, we used circular dichroism spectroscopy (CD) to measure the peaks of the alpha helix structures (208 and 222 nm) and beta sheet (218 nm). Our results revealed alterations in the secondary structure of G-actin upon 12 h of incubation with quantum dots (Figure 5C). At a lower concentration at 10 nM of CdSe/ZnS QDs, the effect of QDs on actin’s secondary structure fluctuated, suggesting that this concentration was not sufficient enough to cause significant changes to G-actin’s secondary structure (Table 2). However, at the concentrations of 25 nM, 50 nM, and 100 nM, the impact of CdSe/ZnS QDs on G-actin was more significant and consistent (Table 2).

## 3. Discussion

The present study reveals, for the first time to our knowledge, via a series of biochemical experiments, that CdSe/ZnS QDs induce quenching of the intrinsic fluorescence of G-actin through static interaction by the formation of QD–actin complexes. This finding is novel, as it suggests that quantum dots can bind and alter the structure of intracellular proteins. It is well known that G-actin is an essential building block for the actin cytoskeleton in cells [51,71]. The process of actin assembly and disassembly is pivotal for the maintenance of cell structures and regulates several processes such as vesicular trafficking and transcription [51]. As protein structure is crucial to the functions and properties of proteins [50], a change in the structure of G-actin (Figure 5C) may impair its ability to participate in F-actin formation, which is of great interest to be investigated. Furthermore, actin is also one of the most abundant proteins in the cells that participates in many necessary protein–protein interactions [51,72]. The binding of G-actin to CdSe/ZnS may limit the G-actin supply necessary for many cellular processes. In our article previously published in 2023, we reported the abnormal appearance of the actin cytoskeleton in *Saccharomyces cerevisiae* yeast upon treatment of CdSe/ZnS QDs. Further, as the expression of actin dynamic regulating proteins, including profilin and coronin, was altered, we proposed that the negative impact of CdSe/ZnS QDs was due to off-balance actin dynamic regulations [39]. In conjunction with this phenomenon, our current data suggest that direct binding of CdSe/ZnS QDs to G-actin may also play a role in the abnormal appearance of the actin cytoskeleton.

Thoroughly assessing the direct impact of QDs on cellular protein, our research provided information on another aspect of QD toxicity. With rising interest in using nanoparticles such as QDs in biomedical applications, the need to evaluate their safety is becoming increasingly important. Many common QD-based biomedical applications such as drug delivery, bioimaging, and cancer detection require QDs to enter the human body and cells [15,44,73,74,75,76,77,78,79]. In these scenarios, understanding the non-specific interactions between QDs and biomolecules such as proteins could help avoid unwanted side effects. The importance of QD-induced protein modification has recently gained attention from researchers. In the last decade, several studies have reported on the interaction between HSA and QDs [57,68,80,81,82]. In 2021, Wang et al. reported that PbS QDs spontaneously interacted with HSA and resulted in the quenching of HSA’s intrinsic fluorescence [82]. They also found that QDs altered the secondary structure of their protein, HSA. Furthermore, they proposed a two-step association process between them: the first step involved hydrophobic interaction and electrostatic interaction of HSA and PbS QDs, and the second step was the formation of a complex carried out by covalent, hydrogen, and van der Waals forces [82]. When compared to our results, QDs seem to use the same fluorescence quenching mode toward both G-actin and HSA. In both cases, QDs spontaneously interact with them and result in protein modifications. The K_a_ value for PbS QDs and HSA interaction is higher (4.73 × 10^7^ L/mol at 298 K, 3.75 × 10^7^ L/mol at 302 K, 2.11 × 10^7^ L/mol at 310 K) compared to K_a_ values in our experiments with G-actin (5.90 × 10^6^ L/mol at 295 K, 4.33 × 10^6^ L/mol at 303 K, 4.11 × 10^6^ L/mol at 301 K) (Table 1), suggesting a higher affinity of QDs toward HAS than G-actin. Interestingly, for the same incubation time of 3 h, the binding ratio of PbS QDs to HSA was reported to be 1:4, while the binding ratios of CdSe/ZnS QDs to G-actin after 3 h incubation were around 1 to 2 and 1 to 2.5. This difference in binding ratios could have resulted from the difference in the experimental setting or the difference in the binding affinity of G-actin to QDs versus HSA to QDs.

Several similar studies have also investigated the binding of HSA or its homologous protein bovine serum albumin (BSA) to different types of QDs. Overall, these studies consistently reported that QDs use static quenching as a fluorescence quenching mode and favor the formation of complexes [57,68,81,82]. The secondary structure of the protein was also altered in this study. In the future, it would be interesting to examine if QDs use similar interaction mechanisms for a wide range of protein types. Additionally, our data identified a number of proteins that interacted with QDs. These are proteins that are associated with transcription, translation, mitochondrial function, and vesicular trafficking. QD treatment has previously been shown to have a negative impact on many cellular processes and components, especially vesicular trafficking and mitochondrial function [39,67,83,84,85,86,87,88,89]. Thus, future research that focuses on the interaction between these proteins with QDs and its impact on protein structure and function may open a novel aspect to learning about QD toxicity.

When considering the interaction of QDs with biomolecules, it is essential to examine the role of functional groups and ligands conjugated on the surfaces of QDs. The functional groups coating QDs are an important component that helps with QD stability [5,90,91,92] and play a key role in the conjugation of desired ligands. Conjugated ligands are useful in applications such as drug delivery, where drugs need to be loaded onto QDs [12,46,93]. Furthermore, new strategies aimed at improving the target specificity of QD-based drug delivery vehicles or QD-based cancer detectors often require the conjugation of a homing peptide with high binding affinity to the target cells [42,45,94,95]. Upon entering the target cells, the release of drugs can be triggered by a low pH in organelles such as the endosomes and lysosomes [12,93]. This drop in pH causes the dissociation of treatments from the delivery vehicle. However, conjugated ligands and homing peptides could also dissociate from QDs, exposing charged functional groups such as carboxyl. Quantum dots with exposed functional groups could then interact with other cellular proteins and cause unforeseen toxicity. Furthermore, it has been shown that QDs are retained in the body for an extended amount of time post-administration [96,97,98]. The loss of homing peptide could likely cause QDs to be uptaken by non-target cells, leading to undesired side effects. In addition, some QDs such as CQDs have been shown to enter the nuclei of cells [99,100]. Thus, the possibility of QDs interacting with DNA should also be explored. For these reasons, understanding the role of QDs’ surface ligand in the interaction between QDs and proteins is needed for the development of safer quantum dots. In our research, we only used CdSe/ZnS QDs conjugated with carboxylic ligands. Thus, we were not able to examine how different ligands affect the binding of QDs to proteins. It would be interesting for future research to investigate the binding of QDs with different ligands with G-actin and other proteins.

## 4. Materials and Methods

### 4.1. CdSe/ZnS QD Characteristics

Water-soluble cadmium selenide zinc sulfide (CdSe/ZnS) QDs with a carboxylic ligand (Cat. #CZW-R-5) were obtained from NN-Labs (Fayetteville, AR, USA). According to the manufacturer, this quantum dot product has an emission peak of 620–635 nm. Previously, Zhang et al. (our lab) measured an air-dry diameter of around 5–10 nm and a hydrodynamic diameter of 20 nm [84]. These numbers were consistent with the manufacturer’s provided information.

### 4.2. Yeast Lysate Preparation

A culture of *Saccharomyces cerevisiae* yeast (strain S288c) was grown in a yeast extract peptone dextrose (YPD) medium overnight to the optical densities of around 0.5 to 0.9 at 600 nm. The culture was then centrifuged, and the pellet was resuspended with 0.5 mL of cold 1× PBS. The cell suspension was then transferred to a microorganism lysing mix with 0.15 mm Garnet (OMNI International, Kennesaw, GA, USA). Next, 500 µL of ice-cold lysis buffer (50 mM HEPES pH 7.3, 200 mM NaCl, 1% Triton X-100, 10 mM imidazole, and 1× protease inhibitor cocktail) was added to the sample. The tubes were placed in a bead beater (OMNI International, Kennesaw, GA, USA) and beaten for 20 s at 5.5 m/s at 4 °C. This beating process was repeated 6 times, with 1 min of ice incubation between each time to prevent protein degradation. The samples were then centrifuged for 15 min at 15,000 rpm at 4 °C. The supernatant containing proteins was collected and dialyzed against a buffer (50 mM Tris-HCl, pH 7.4, 100 mM NaCl, and 5 mM MgCl_2_) overnight at 4 °C to remove impurities.

### 4.3. Proteomic Shotgun Analysis

We created a QD–bead complex by conjugating negatively charged carboxylic ligands on the quantum dots with amine-coated magnetic beads (SKU: MGB-NH2-10-10, Luna Nanotech, Toronto, ON, Canada) using a 1-ethyl-3-[3-dimethylaminopropyl] carbodiimide hydrochloride (EDC) protein crosslinker (Pierce Biotechnology, Rockford, IL, USA). To achieve this, 2 nM of CdSe/ZnS QDs were mixed with 2 mg of anime magnetic beads in the presence of EDC. The mixture was incubated for 6 h at room temperature with gentle agitation. We confirmed the conjugation by visualizing the isolated bead–QD mixture under a fluorescence microscope. Afterward, the conjugated mixture was dialyzed against 1X PBS overnight at 4 °C to remove the reaction’s byproduct.

Three protein lysate samples obtained by a standard bead beating method as stated in Section 4.2 were dialyzed against a binding buffer without glycerol (50 mM Tris HCl, pH 7.4, 10 mM NaCl, and 5 mM MgCl_2_) overnight at 4 °C. Each protein lysate sample was transferred to one of the following solutions: (1) QD-conjugated to COOH-coated magnetic beads; (2) COOH-coated magnetic beads (SKU #MGB-COOH-10-10); or (3) NH_2_-coated magnetic beads (SKU #MGB-NH2-10-10). The mixture was allowed to bind in the binding buffer (Invitrogen, Vilnius, Lithuania) at 4 °C overnight. Unbound proteins were washed away using cold 1× PBS. The 1× SDS loading buffer (2% SDS, 100 mM Tris-HCl, pH 6.8) was added for 2 h to cause protein dissociation from the QD–magnetic bead complex. Dissociated proteins were then dialyzed in 1× PBS overnight at 4 °C. Samples were centrifuged and a supernatant containing proteins was obtained. Protein samples were lyophilized for 2 days to remove excess liquid. An amount of 100 µL of 1× PBS was added to the protein sample, and protein concentration was measured by using a BSA standard curve. A total of 50 µL of protein was shipped in dry ice to Creative Proteomic (Shirley, NY, USA) for shotgun proteomics. After arrival, samples were loaded into an SDS-PAGE gel (12% separating gels) and run at 80 kV for 30 min, then 120 kV for an hour to fully separate protein bands. Protein bands were stained and cut into 1 mm^3^ cubes and transferred into a microcentrifuge tube. Next, proteins were removed from the gel slice by trypsin digestion. The supernatant containing the resulting peptides was lyophilized into powder. The peptide samples were resuspended in 20 μL of 0.1% formic acid before an LC-MS/MS analysis. The samples were assessed by an UltiMate 3000 nano UHPLC system and a Q Exactive HF mass spectrometer (Thermo Fisher Scientific, Waltham, MA, USA). Raw MS data files were analyzed against the *Saccharomyces cerevisiae* protein database in the MaxQuant program (1.6.2.6). The experiment was conducted in a triplicated manner. Diagrams of identified proteins and impacted processes were created with BioRender.com.

### 4.4. Actin Preparation

Lyophilized rabbit skeletal muscle actin powder (Cat. #AKL99-C) was obtained from Cytoskeleton (Denver, CO, USA). First, the protein powder was solubilized with 100 µL of pure water according to the instructions from the manufacturer. The solubilized actin (10 mg/mL) was aliquoted into several vials and snap-frozen in liquid nitrogen per the manufacturer’s protocol. The actin vials were then stored at −80 °C until use.

Before each experiment, the aliquoted actin vials were quickly thawed by briefly being held in the palm of the hand then immediately placed on ice and diluted with G-buffer (5 mM Tris HCl, pH 8, 0.2 mM CaCl_2_, and 0.2 mM ATP) to bring the concentration down to 0.4 mg/mL. Then, the diluted actin vials were incubated for 45 min to an hour on ice according to the manufacturer’s instructions. Finally, the G-actin was centrifuged for 10 min at 13,000 rpm at 4 °C. The supernatant was immediately used for experimental purposes. Any leftover diluted actin was discarded, and a fresh actin batch was made before each experiment.

### 4.5. Native Gel Electrophoresis

Before each experiment, Mini-PROTEAN precast polyacrylamide gels (Bio-Rad, Hercules, CA, USA) were pre-run in a native gel buffer (25 mM Tris-HCl pH 8, 194 mM glycine, 0.5 mM CaCl_2_, 0.2 mM DTT, and 0.2 mM ATP) for one hour at 4 °C. Samples for fixed actin and increasing concentrations of QD assays were then created by mixing 2 µM of G-actin with 0.025 µM, 0.05 µM, 0.1 µM, 0.2 µM, 0.4 µM, 0.8 µM, or 1.6 µM of QDs. Samples for fixed quantum dots and increased concentrations of actin were created by mixing 0.2 µM of QDs with 0.4 µM, 0.8 µM, 1.2 µM, 1.6 µM, 2 µM, 4 µM, or 8 µM of G-actin. Additional G-buffer was added to each sample to yield a total volume of 10 µL. The samples were then incubated for 3 h at room temperature. After incubation time, 1 µL of 6× loading dye and 1 µL of glycerol were mixed into each of the samples. Then, the samples were carefully loaded into each well of the pre-run native gel submerged in fresh native gel buffer. The gels were then run for 50 min at 190 volts at room temperature. Afterward, the gels were stained in Coomassie blue dye for 30 min with gentle agitation and washed with native gel destaining solution for about an hour. Images of the gels were then taken. The band intensity on gels was measured using ImageJ version 1.53 t and the data were analyzed and graphed by Prism GraphPad 9.

### 4.6. Fluorometer-Based Actin Fluorescence Quenching

A fixed concentration of 2 µM of G-actin was incubated with different concentrations of QDs (4.12 nM, 10.6 nM, and 20.6 nM) for 3 h in 295 K, 303 K, or 310 K. Afterward, the intrinsic fluorescence of G-actin was measured using a PTI spectrofluorometer (PTI Photon Technology International, Birmingham, NJ, USA) with an excitation wavelength of 280 nm and emission range from 300 to 420 nm. The excitation bandwidth was set to 8 nm and the emission bandwidth was set to 10 nm. The obtained data were then graphed using Prism GraphPad version 9.5.0.

### 4.7. Cd^2+^ Ion Leakage Detection

The release of free cadmium ions (Cd^2+^) from QDs was tested in both deionized (DI) water (18.2 MΩ/cm) and G-buffer at a final QD concentration of 8 ppm (8 mg/L) using an anodic squarewave voltammetry (ASWV) method. For the QDs in G-buffer, samples were tested as freshly prepared which was denoted as day 0, and the Cd^2+^ release was measured again after 14 days of incubation (day 14) at 4 °C. For the QDs in DI water samples, ASWV tests were performed at 33 days of incubation (day 33).

Prior to electrochemical measurement, sodium acetate/potassium nitrate buffer (pH 5) was added to all sample solutions to a final concentration of 0.01 M in order to maintain constant pH and ionic conductivity for ASWV testing. The final pH of DI water samples and G-buffer samples were 5 and 8, respectively.

Standard solutions of 50 ppb, 500 ppb, and 1 ppm Cd^2+^ were prepared in both 0.01 M sodium acetate/potassium nitrate buffer (pH 5) and 0.01 M G-buffer (with 0.01 M sodium acetate/potassium nitrate) to validate the ASWV method. Additionally, solutions of 0.01 M sodium acetate/potassium nitrate buffer and G-buffer without QDs or Cd^2+^ ions were prepared and tested as control samples.

To measure the level of free Cd^2+^ ions in solution, anodic squarewave voltammetry (ASWV) experiments were performed using a Gamry Interface 1010E Potentiostat (Gamry Instruments Inc., Warminster, PA, USA). A three-electrode system consisting of a platinum wire counter electrode, Ag/AgCl reference electrode, and glassy carbon working electrode (3 mm diameter) was used. Potentials were applied from −1.0 V to +0.2 V (deposition potential at −1.0 V), with a 60 s accumulation time, 60 s equilibration time, frequency of 5 Hz, and pulse size of 25 mV. The method detection limits for Cd^2+^ ions in the DI water samples and G-buffer samples were around 50 ppb and 100 ppb, respectively. All data were collected in triplicate at room temperature.

### 4.8. Ultraviolet–Visible (UV–Vis) Absorption Spectroscopy

Similar to the fluorometer experiment, a fixed concentration of 0.2 µM G-actin was added to 2 nM of QDs and incubated at room temperature (295 K) for 3 h. The absorbance of our samples (QDs alone, actin alone, and actin with QDs) was scanned from 190 to 500 nm with a UV-2101PC UV–Vis spectrophotometer (Shimadzu, Columbia, MD, USA). The experiments were conducted in a triplicated manner and the data were graphed using GraphPad Prism 9.

### 4.9. Dynamic Light Scattering (DLS)

Prior to performing DLS, the G-actin samples at 0.4 mg/mL were ultracentrifuged with an Optimal MAX Ultracentrifuge (Beckman Coulter, Brea, CA, USA) at 100,000× *g* with a TLA-120.2 rotor at 4 °C for 45 min. The supernatant of the solution was used to carry out further experimental work.

A fixed concentration of G-actin (2 µM) was mixed with different concentrations of quantum dots (4.86 nM, 19.44 nM, 38.8 nM, and 48.60 nM) at room temperature (295 K). The diameters of actin alone, CdSe/ZnS QDs alone, and actin mixed with different concentrations of quantum dots were measured using a NanoBrook Omni Dynamic Light Scattering Particle Sizer (Brookhaven, Fresno, GA, USA). All samples were run three times. The data were processed and graphed by GraphPad Prism 9.

### 4.10. Circular Dichroism

A fixed concentration of 10 µM of G-actin was mixed with different concentrations of QDs (10 nM, 25 nM, 50 nM, and 100 nM) and incubated for 3 h and 12 h at room temperature (295 K). Subsequently, the CD spectra of our samples (actin alone and actin mixed with different concentrations of QDs) were measured with a J-815 CD Spectrometer (JASCO, Oklahoma, OK, USA) at 295 K. Each sample measurement was an average of three repeats with the scanning range set to 190–260 nm and a scanning speed of 50 nm/min. The obtained data were then graphed using GraphPad Prism 9.

## Figures and Tables

**Figure 1 ijms-24-14760-f001:**
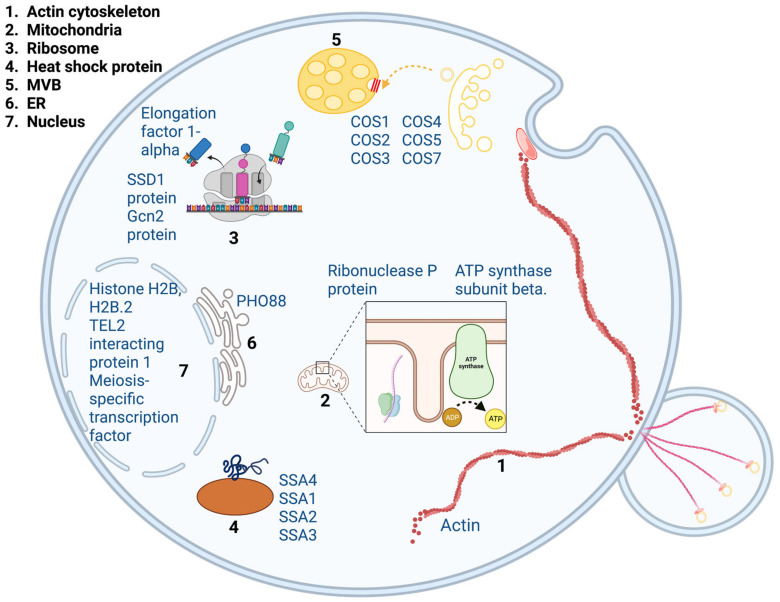
Structures and processes of QD binding proteins including the actin cytoskeleton, mitochondrial structure and function, translation (ribosome), transcription, inorganic transport, vesicular trafficking, and heat shock proteins. Protein names are in blue.

**Figure 2 ijms-24-14760-f002:**
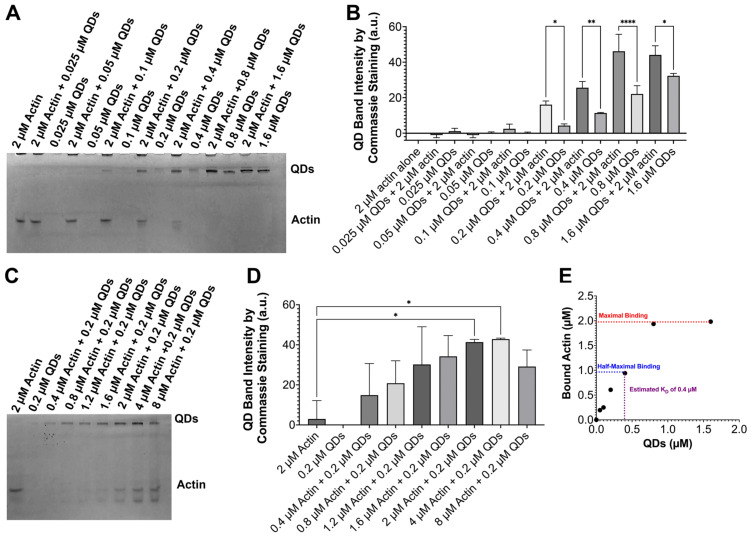
Native polyacrylamide gel electrophoresis to determine the binding ratio of QDs to actin and the K_D_. The top band represents QD bands and the bottom bands are free G-actin bands. (**A**) Gel with fixed 2 µM of actin incubated with increasing concentrations of QDs. (**B**) Graph quantifying the intensity of the QD bands by Coomassie staining in gel A. (**C**) Gel image of fixed 0.2 µM of QDs incubated with increasing concentrations of actin. (**D**) Graph quantifying the intensity of the QD bands by Coomassie staining in gel C. (**E**) K_D_ estimation based on the method outlined by Chambers et al. [56]. The estimated K_D_ values are equal to the concentration of QDs that bind to the half-maximum binding of G-actin. * *p* < 0.05, ** *p* < 0.01, **** *p* < 0.0001.

**Figure 3 ijms-24-14760-f003:**
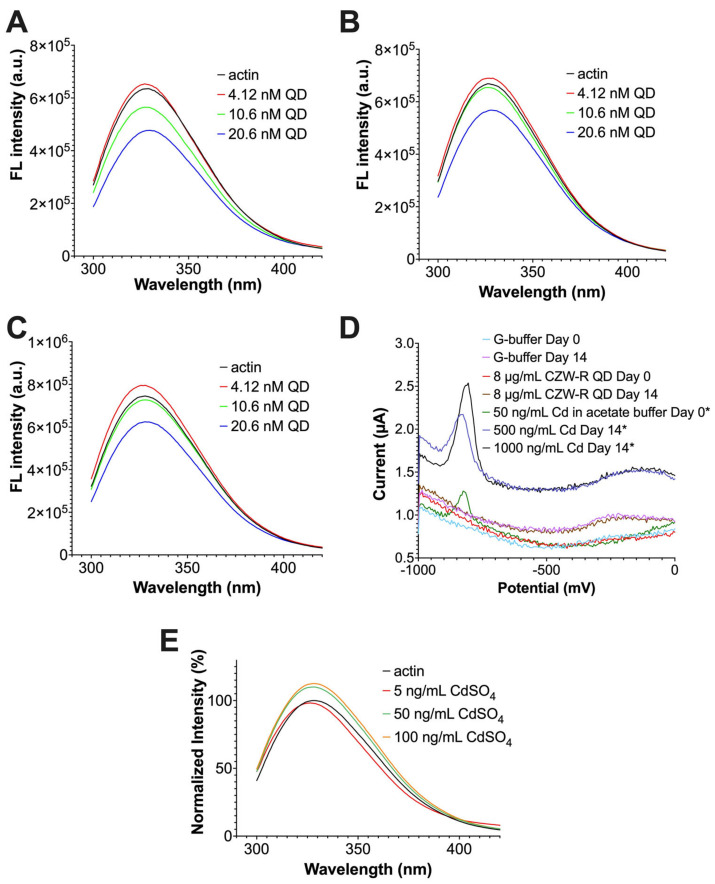
The quenching of G-actin intrinsic fluorescence by CdSe/ZnS QDs at different temperatures as well as cadmium ions in the form of CdSO_4_ at 295 K. (**A**) The intrinsic fluorescence of G-actin in the presence of CdSe/ZnS QDs at 295 K. (**B**) The intrinsic fluorescence of G-actin in the presence of CdSe/ZnS QDs at 303 K. (**C**) The intrinsic fluorescence of G-actin in the presence of CdSe/ZnS QDs at 310 K. (**D**) Detected Cd^2+^ leakage level from 8 µg/mL (8 ppm) of CdSe/ZnS QDs after 0 and 14 days. * 50, 500, and 1000 ng/mL or ppb of cadmium ions were used as a positive control. (**E**) The intrinsic fluorescence of G-actin in the presence of cadmium ions (CdSO_4_) at 295 K.

**Figure 4 ijms-24-14760-f004:**
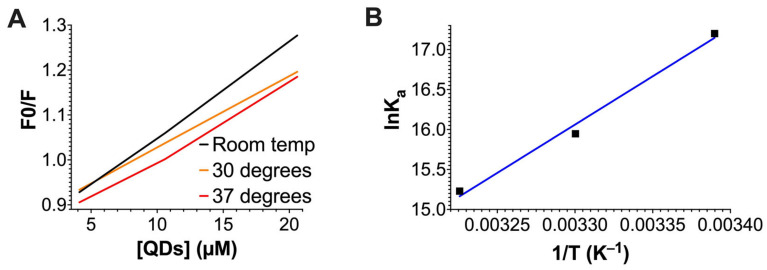
An analysis of the quenching of actin’s intrinsic fluorescence by CdSe/ZnS QDs using (**A**) Stern–Volmer’s plot. The slope of each line represents the K_sv_ values at different temperatures and (**B**) the inverse relationship between K_a_ (expressed in lnK_a_) and temperatures (expressed in 1/T).

**Figure 5 ijms-24-14760-f005:**
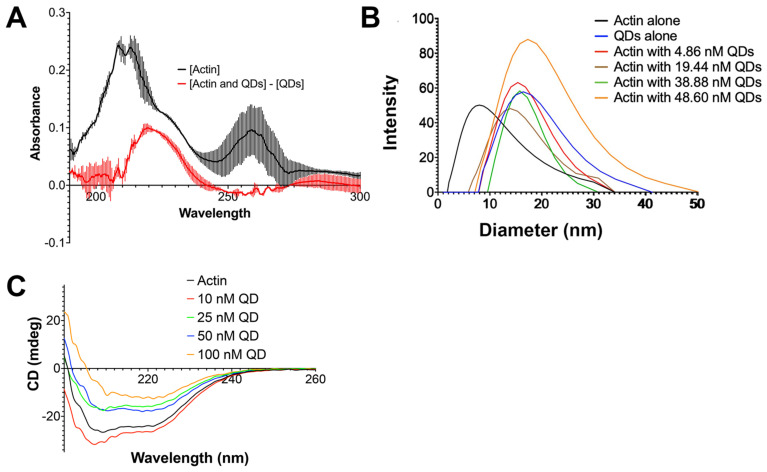
(**A**) A plot of the absorbance curve for G-actin incubated with quantum dots and G-actin alone. (**B**) Dynamic light scattering graph showing the shift in diameter of actin in the presence of quantum dots. (**C**) Circular dichroism graph measuring the secondary structure of G-actin at different concentrations of quantum dots.

**Table 1 ijms-24-14760-t001:** Summary of the thermodynamic parameters of CdSe/ZnS QDs and G-actin. T is the temperature in Kelvin, K_SV_ is the Stern–Volmer constant, K_a_ is the association constant, ∆G is the change in Gibbs free energy, ∆H stands for enthalpy change, and ∆S stands for entropy change.

T (K)	K_sv_ (L mol^−1^)	R^2^	K_a_ (L/mol^−1^)	R^2^	∆G (kJ mol^−1^)	∆H (kJ mol^−1^)	∆S (J mol^−1^ K^−1^)
310	8.98 × 10^6^	0.969	4.11 × 10^6^ ± 3.91× 10^5^	0.826	−39.25 ± 0.22	−18.69 ± 4.5	65.99 ± 14.5
303	9.52 × 10^6^	0.986	4.33 × 10^6^ ± 1.14 × 10^6^	0.872	−38.49 ± 0.46		
295	1.34 × 10^7^	0.980	5.90 × 10^6^ ± 1.85 × 10^6^	0.808	−38.24 ± 0.88		

**Table 2 ijms-24-14760-t002:** Percent change of the G-actin secondary structure peaks at the wavelength of 208 nm, 222 nm, and 218 nm in the presence of 10 nM, 25 nM, 50 nM, and 100 nM of CdSe/ZnS QDs compared to the G-actin control. Peaks at 222 nm and 208 nm represent the alpha helix structure and peaks at 218 nm represent the beta sheet structure.

Percent Change				
Wavelength	10 nM	25 nM	50 nM	100 nM
222 nm	34.1 ± 30.8	56.1 ± 17.2	47.7 ± 19.9	69.4 ± 1.5
208 nm	30.4 ± 32.3	53.8 ± 13.8	54.3 ± 15.2	88.8 ± 2.0
218 nm	32.9 ± 30.9	55.0 ± 17.7	47.3 ± 19.9	71.3 ± 1.9

## Data Availability

Not applicable.

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
