# Peer review of "Interactions between Quantum Dots and G-Actin"

_ijms, 2023, doi:10.3390/ijms241914760_

Round 1

Reviewer 1 Report

The reviewed paper elucidates an interaction between important protein actin and quantum dots. From the point of importance and novelty the topic is highly relevant and current. Analytical techniques for the investigation were well chosen and the results are quite interesting. Primarily, it was shown by electrophoresis that QD-actin complex in ratio 1:2-2.5 is spontaneously formed. The question is why subsequent experiments have been performed with such a different QD:actin ratio – such as, in UV-vis 1:1000, in DLS approximately 1:40, in CD spectroscopy 1:100. Is it expectable to see significantly strong signals of complex in such an excess of signal of unreacted actin?

Furthermore, SWV, although powerful method, doesn´t seem to be the best choice under the given conditions. Pristine carbon electrode may not provide enough sensitivity and low enough limit of detection. Especially at pH 8, it is known that metal ions are best detected at much lower pH. It is also of question whether to use a method with LOD of 50 ppm when the original sample contained only 8 ppm of QDs. It is also not clear what does XXng/mL Cd Day 14 in the figure 3D mean.

Further minor issues:

-          more detailed discussion about interaction of proteins and QDs should be provided in the introduction than in the discussion section.

-          what exactly in results in the section 3.3 justified the claim that “..QDs…caused the alteration in the position of tryptophan.”? (line 306-307). Why not phenylalanine?

-          line 321 – should be probably Figure 3E, not 4E

-          caption of figure 3 should be rewritten such as to allow “self understanding” of the figure even without the read of the text. In panel D it should be explained what is Cd Day 14. In panel E the curves are probably not for pure CdSO4, but for actin with CdSO4 and so on.

-          What is the different between evaluation using Stern-Volmer equation and the clear evaluation given at lines 307-311?

-          Figure 5B – it is difficult to match colors of the curves with colors in the legend. Furthermore, the curve for pristine QDs seems to be absent.

-          In the table 2 there are red values for “turns” and “unordered”, but in the text it is said that their proportion has increased. Furthermore, questions may be raised on reliability of results where standard deviation is greater than the value itself. May such data scattering be connected to great excess of actin in the sample solutions? (according to the Material and methods section QD:actin ratio is 1:100). The achieved CD spectroscopy results should be also more interpreted, not only described. Is it possible to estimate at which site of the protein the structure change occurs? Is it possible to draw a scheme of the observed structural change?

Conclusion: although the study is quite original and interesting, it should be revised according to the comments above.

Author Response

Please find the attached rebuttal letter

Reviewer 2 Report

In this study the authors investigate the effect that cadmium-based quantum dots have on G-actin, an important protein, and showed that the protein has interactions with the quantum dots. In general, the study seems very thorough and well-done, and I have no outstanding complaints with the study, and I believe it is ready to be accepted for publication.  I have a few thoughts of what might improve the manuscript.

1. The quantum dots used herein were purchased commercially, and no characterization was done. I think it would be nice to add some brief characterization data (UV-vis, luminescence, TEM) perhaps to some supplemental material.

2. The thermodynamics data is presented oddly in Table 1. Using the van't Hoff equation, the slope and intercept can be used to calculate the enthalpy and entropy. However, I do not understand why three values are displayed for the entropy of the reaction, when there should only be one fit value (as seen for the enthalpy). Additionally, the units of kJ/mol K are not typical for entropy values, and J/mol K should be used. However, the authors could instead report TdS for the given temperature value (in units of kJ/mol) which is probably the best representation of entropy data in this reviewer's opinion (as it allows direct comparison with enthalpy information) and if this was done there would be three different values for the three different temperatures. If only the entropy is reported, it should only be reported once, and probably not alongside the temperature data. I would report the enthalpy and entropy calculated simply in the manuscript if the authors choose to go this route.

3. There are an unfortunate lack of error bars, particularly in Table 1. The authors should attempt to quantify the error in the reported parameters (easily done for the slope and intercept used for the enthalpy and entropy values) and an estimated error for the equilibrium constants would be appreciated as well.

Author Response

Please find the attached rebuttal letter.

Round 2

Reviewer 1 Report

The revision has been done as requested and the paper can now be accepted.